# Excitonic-Vibrational Interaction at 2D Material/Organic Molecule Interfaces Studied by Time-Resolved Sum Frequency Generation

**DOI:** 10.3390/nano14231892

**Published:** 2024-11-25

**Authors:** Huiling Chen, Yu Lian, Tao Zhou, Hui Li, Jiashi Li, Xinyi Liu, Yuan Huang, Wei-Tao Liu

**Affiliations:** 1Physics Department, State Key Laboratory of Surface Physics, Key Laboratory of Micro and Nano Photonic Structures [Ministry of Education (MOE)], Fudan University, Shanghai 200433, China; 2School of Integrated Circuits and Electronics, MIIT Key Laboratory for Low-Dimensional Quantum Structure and Devices, Beijing Institute of Technology, Beijing 100081, China

**Keywords:** 2D hybrid heterostructure, TMDC/organic interface, sum-frequency generation, ultrafast dynamics, surface field enhancement, excitonic–vibronic coupling

## Abstract

The hybrid heterostructures formed between two-dimensional (2D) materials and organic molecules have gained great interest for their potential applications in advanced photonic and optoelectronic devices, such as solar cells and biosensors. Characterizing the interfacial structure and dynamic properties at the molecular level is essential for realizing such applications. Here, we report a time-resolved sum-frequency generation (TR-SFG) approach to investigate the hybrid structure of polymethyl methacrylate (PMMA) molecules and 2D transition metal dichalcogenides (TMDCs). By utilizing both infrared and visible light, TR-SFG can provide surface-specific information about both molecular vibrations and electronic transitions simultaneously. Our setup employed a Bragg grating for generating both a narrowband probe and an ultrafast pump pulse, along with a synchronized beam chopper and Galvo mirror combination for real-time spectral normalization, which can be readily incorporated into standard SFG setups. Applying this technique to the TMDC/PMMA interfaces yielded structural information regarding PMMA side chains and dynamic responses of both PMMA vibrational modes and TMDC excitonic transitions. We further observed a prominent enhancement effect of the PMMA vibrational SF amplitude for about 10 times upon the resonance with TMDC excitonic transition. These findings lay a foundation for further investigation into interactions at the 2D material/organic molecule interfaces.

## 1. Introduction

Two-dimensional (2D) materials, such as graphene, black phosphorus, and transition metal dichalcogenides (TMDCs), have rich and unique optical and electrical properties that drive innovations in related fields [1]. When combined with organic molecules in hybrid heterostructures, they provide a compelling platform that combines favorable properties of both, which is of great interest to applications including solar cells, photovoltaics, and biosensors [2,3,4]. Studies have shown that these interfaces can facilitate rapid charge transfer and long-lived charge-separated states, along with resonant energy transfer between the organic molecules and neighboring 2D layers that enhances light–matter interactions [5,6,7,8,9]. However, our understanding of these processes remains limited, in particular at the molecular level. The main challenge lies in that most spectroscopy [static and time-resolved (TR)] techniques are not specifically tailored for surfaces and interfaces. Although 2D layers are atomically thin, organic molecules are usually deposited as thin films, with thickness typically ranging from a few tens to a few hundreds of nanometers [10]. The molecules interact differently with 2D materials at the interface compared with those within the film, necessitating the use of surface-specific techniques for accurate analysis.

Sum-frequency generation (SFG) is a highly surface-specific nonlinear optical technique, applicable even to buried interfaces that are accessible to photons [11]. It is a powerful tool for studying molecular interactions at surfaces and interfaces. Figure 1a depicts the quantum transition pathways of an SFG process for vibrational spectroscopy: it includes an infrared (IR) excitation of the vibrational coherence (purple arrow), followed by an anti-Stokes Raman stage involving a one-photon transition to upper levels (orange arrow), then back to the ground state (blue arrow). Since the infrared and Raman selection rules are mutually exclusive in centrosymmetric systems, SFG is allowed only when the inversion symmetry is broken, such as on surfaces and interfaces [11]. By utilizing the IR resonance of molecules and visible light for electronic excitation, SFG enables the simultaneous probing of both transitions at interfaces for studying the vibronic interaction [12,13]. Employing ultrashort laser pulses and pump-probe schemes (Figure 1b), we can further utilize time-resolved SFG (TR-SFG) to elucidate the interfacial dynamic processes. Previous TR-SFG studies have been primarily used to study the vibrational dynamics of molecular adsorbates [14,15]. The properties of heterostructures between 2D materials and organic molecules still await in-depth investigations.

Here, we report a TR-SFG experimental approach to study interfaces between organic molecules and 2D materials. The setup was designed for easy adaptation from a standard broadband SF spectroscopy configuration. We employed a single Bragg grating to simultaneously generate the narrowband (~0.5 nm) probe beam and the ultrafast (FWHM ~95 fs) pump beam, which largely simplified the beam geometry. We further adopted an electronic synchronization scheme incorporating a beam chopper and a Galvo mirror, allowing real-time normalization of the SF spectra both with and without the pump excitation [16]. As a demonstration, we examined the interfaces of MoS_2_/PMMA (polymethyl methacrylate) and WS_2_/PMMA, with PMMA being a common coating material on 2D layers [17]. Using SFG, we observed that methyl groups [(O)CH3] on PMMA ester side chains in contact with TMDC layers tilted away from the surface normal, contrasting with their orientations at the PMMA/air interface. Notably, we detected a significant enhancement in the vibrational SF response from PMMA upon the simultaneous excitation of excitonic transitions in MoS_2_. This enhancement was absent in WS_2_, where the exciton energy did not align with the PMMA vibrational SFG. Combined with the TR-SFG data, we attribute this enhancement to arise from the resonance between PMMA vibrational SFG and MoS_2_ excitons, which are coherently excited by the same SFG process.

## 2. Materials and Methods

### 2.1. The Basic Principle of SFG

The basic theory of SFG is described elsewhere [11]. Briefly, in our SF spectroscopy experiment, an infrared (IR) and a near-IR (NIR) laser beam were overlapped spatially and temporally at the surface/interface, generating a sum-frequency (SF) signal at the sum of two input frequencies that could be detected along coherent directions. When the IR beam was near a vibrational mode, the SF signal was resonantly enhanced, which manifested in the SF spectrum as a resonance peak. The SF signal intensity was proportional to the square of the product of the local electric fields and the second-order effective nonlinear optical susceptibility $\chi_{eff}^{\left(2\right)}$ [11]:(1)$$S(\omega_{SF}=\omega_{NIR}+\omega_{IR})\propto {\left|\chi_{eff}^{\left(2\right)}\right|}^{2},$$
$$\chi_{eff}^{\left(2\right)}=\left[{\hat{e}}_{SF}\cdot {\overset{⃡}{L}}_{SF}\right]\cdot {\overset{⃡}{\chi }}_{tot}^{\left(2\right)}:\left[{\hat{e}}_{NIR}\cdot {\overset{⃡}{L}}_{NIR}\right]\left[{\hat{e}}_{IR}\cdot {\overset{⃡}{L}}_{IR}\right],$$
$${\overset{⃡}{\chi }}_{tot}^{\left(2\right)}={\overset{⃡}{\chi }}_{NR}^{\left(2\right)}+{\overset{⃡}{\chi }}_{R}^{\left(2\right)},$$
where ${\hat{e}}_{i}$ and ${\overset{⃡}{L}}_{i}$ are the unit polarization vector and local field correction factor at $\omega_{i}$, respectively. The total susceptibility, ${\overset{⃡}{\chi }}_{tot}^{\left(2\right)}$, is the sum of the nonresonant background, ${\overset{⃡}{\chi }}_{NR}^{\left(2\right)}$, and the discrete resonance contributions, ${\overset{⃡}{\chi }}_{R}^{\left(2\right)}$. When the IR frequency $\omega_{IR}$ is near molecular resonances, ${\overset{⃡}{\chi }}_{R}^{\left(2\right)}$ can be expressed as
(2)$${\overset{⃡}{\chi }}_{R}^{\left(2\right)}=\sum_{q}\frac{{\overset{⃡}{A}}_{q}^{\left(2\right)}}{\omega_{IR}-\omega_{q}+i\Gamma_{q}},$$
with ${\overset{⃡}{A}}_{q}^{\left(2\right)}$, $\omega_{q}$, and $\Gamma_{q}$ denoting the amplitude, resonant frequency, and damping coefficient of the $q^{th}$ resonant mode of the contributing species, respectively.

### 2.2. Sample Preparation

Undoped gallium arsenide (GaAs) (100) wafers (Shanghai Institute of Technical Physics) were used as reference sample in our experiments. TMDC samples were prepared using the modified mechanical exfoliation method for 2D materials [18]. This method could yield large-area TMDC monolayer flakes [18] with lateral sizes of 200~300 μm, as used in our experiment. Prior to exfoliating the TMDC monolayers, the fused silica substrate was ultrasonicated sequentially in acetone, 2-propanol, and deionized water and then subjected to oxygen plasma to remove contaminants from the surface. The MoS_2_ and WS_2_ flakes were peeled off from bulk crystals using adhesive tape and then brought into contact with the substrate. The silica substrate with the attached tape was then annealed for 2 to 5 min at 100 °C in a glove box with N_2_/He atmosphere on a hot plate. After cooling to room temperature, the adhesive tape was removed to complete the exfoliation. The PMMA was then spin-coated onto TMDC monolayers at 3000 rpm for 1 min. The thickness of the PMMA film was estimated to be ~250 nm according to the FTIR measurement.

### 2.3. Optical Experimental Setup

#### 2.3.1. The Generation of Laser Pulses

The experimental setup of TR-SFG is schematically depicted in Figure 1c [19]. The laser system consisted of a commercial Ti:sapphire amplifier (Solstice Ace, Spectra Physics, Santa Clara, CA, USA) that generated pulses centered at 800 nm, with a pulse energy of ~3.6 mJ, a bandwidth of 50 nm, and a pulse duration of 86 fs, operating at a repetition rate of 2 kHz. The amplifier output was first split into two parts by a beam splitter. Thirty-five percent of the total energy was used to pump a commercial optical parametric amplifier (OPA, TOPAS-Prime, Light Conversion, Vilnius, Lithuania) coupled to a noncollinear difference frequency generation (NDFG) stage, generating a broadband IR output tunable between 2.6 µm and 15 µm, with a pulse energy of about 1 to 30 µJ (depending on the IR wavelength). The typical FWHM and pulse duration of the IR pulse were ~400 cm^−1^ and ~100 fs.

The remaining 65% of the amplifier output was used to generate the pump and probe beams for TR-SFG. Instead of the more widely employed 4-*f* configuration with grating pairs [16,20], we used a single optical Bragg grating (BPF-800, OptiGrate, Oviedo, FL, USA). When the Bragg condition was met, the reflected beam was spectrally narrowed down to an FWHM of ~0.5 nm (~8 cm^−1^) centered at 800 nm, which was used as the NIR input for SFG experiments (probe pulse). Meanwhile, instead of being dumped, the transmitted beam through the Bragg grating was directly utilized as the pump beam for the TR-SFG measurement. The typical energy of the NIR probe pulse was ~25 µJ, and that of the transmitted pump pulse was ~2.2 mJ. We determined the pulse duration of the fs-NIR pump pulse to be about 95 fs using an SHG cross-correlator (Figure 1d). Since the reflected bandwidth (~0.5 nm) was much narrower than the original bandwidth (~50 nm), the duration of the transmitted NIR pulse (95 fs) was only slightly longer than the original one (86 fs). An additional OPA/DFG system can be readily inserted into the pump beam path to tune the wavelength for future multi-color TR-SFG measurements.

#### 2.3.2. TR-SFG Experimental Geometry

As displayed in the bottom of Figure 1c, the NIR pump, NIR probe, and IR beams were kept in the same vertical plane of incidence and focused to beam spots of about ~1200, 500, and 420 µm in diameters on the sample, respectively. The narrowband NIR and broadband IR probe beams overlapped spatially and temporally (via Delay 1) on the sample surface at incident angles of ~45° and ~62° with respect to the surface normal, respectively. They generated the broadband SF signals mentioned below in the paper if not specified otherwise. The NIR pump beam also spatially overlapped with probe beams at an incident angle of ~27.6°. A motorized translational stage (MFA-CC, Newport, Delay 2) was used to control the time delay between the pump and probe pulses. The time zero was monitored using the SFG cross-correlation signal of the NIR pump and IR probe from the reference sample [21]. A pair of half-wave plates and a thin film polarizer (Glan-laser polarizer) were used to tune the fluence and polarization of each NIR beam. In most measurements, we used about 25~50 µJ, ~5 µJ, and ~10 µJ for the NIR pump, NIR probe, and IR pulses, respectively.

The SF signal generated from the sample surface was collected along the reflection direction and collimated by a pair of lenses with focus lengths of 200 and 75 mm, then directed to a spectrograph (Princeton Instruments, HRS 300 mm, Trenton, NJ, USA) coupled with a liquid nitrogen-cooled CCD camera (Princeton Instruments, PyLoN: 400BRX, 1340 × 400 pixels, 26.8 mm × 8 mm image area). Data acquisition for both static (without pump) and transient (with pump) SF signals was performed using the Lightfield software (Princeton Instruments) and a self-compiled LabVIEW program. Before entering into the spectrograph, the fundamental beams and stray light were spectrally and spatially filtered out from the signal. A Glan-laser analyzer was used for checking the polarization state of the SF signal. For all measurements presented in this study, the polarization combination was SSP, representing the S-polarized SFG output, S-polarized NIR input, and P-polarized input IR. The polarization of the NIR pump beam was also set to be S-polarized. All SF spectra were normalized to those from an *α*-quartz (0001) sample taken under the same conditions [22]. All measurements were performed at room temperature under the ambient atmospheric pressure.

#### 2.3.3. Detection and Data Acquisition

To improve the signal-to-noise-ratio (SNR) of TR measurements, we used a lock-in detection strategy to achieve real-time normalization between SF spectra with and without a pump on the same sample spot, similar to that described in Ref. [16]. This was achieved by using a phase-locked optical beam chopper (Thorlabs, MC2000B, Newton, NJ, USA), which periodically switched on/off the pump beam every ~0.05 s (corresponding to ~100 pulses) (Figure 1c). We then spatially separated the pump-on and pump-off SF spectra using a single-axis Galvo mirror (Thorlabs, GVS001), oscillating with a small angle of ~0.12° about the mirror axis (Figure 1c). The SF spectra were then imaged into two stripes with a vertical separation of ~1.2 mm on the CCD chip (total chip size being 8.0 mm for 400 pixels) (Figure 1e). The oscillation of the Galvo mirror was locked to the beam chopper using two synchronous 20 Hz square-wave electrical pulse trains from a dual-channel function generator (RIGOL, DG1022U, Suzhou, China), with independently adjustable phase and amplitude. By fine tuning the synchronizing electrical pulse trains, we ensured that the two vertically binned strips on the CCD chip corresponded to the pump-on and pump-off SF spectra. A typical CCD image of SF spectra with the chopper–Galvo system in action is shown in Figure 1e, where the upper and lower traces correspond to the pump-on and pump-off SF spectra, respectively. As a result, the pump-induced change in the TR-SFG signal can be obtained as [23]
(3)$$\frac{∆I}{I}(\omega ,\;t)=\frac{I_{on}(\omega ,\;t)-I_{off}(\omega ,\;t)}{I_{off}(\omega ,\;t)},$$
where $I_{on}(\omega ,\;t)$ and $I_{off}(\omega ,\;t)$ are the SF spectra with and without pump, respectively, with *t* being the time delay between the pump and probe. Since the pump-on and pump-off spectra were acquired during the same period of time, this measurement scheme could minimize the influence from long-term fluctuation and drift of the laser system and measurement stage, thus improving the SNR.

## 3. Results and Discussion

### 3.1. TR-SFG Experiments of GaAs

We first tested our TR-SFG setup using a GaAs (100) wafer. Having a non-centrosymmetric *T_d_* point group, GaAs is known to have a strong SFG response across the IR to visible frequency range and has been extensively studied for its ultrafast dynamic properties [23,24,25]. We therefore employed GaAs to calibrate and characterize our TR-SFG setup. In the measurement, the beam incident plane was kept at an angle of 45° from the GaAs [001] axis, the NIR probe was centered at 1.55 eV (800 nm), and the broadband IR was between 0.3 and 0.4 eV. Since the direct bandgap of GaAs is 1.42 eV at room temperature [24], the transition can be resonantly excited by the NIR pulse. Figure 2a displays the static SF spectrum from GaAs, which showed a broadband SF response between 1.85 and 1.95 eV. The spectrum exhibited two resonant features centered near *ω_IR_* = 0.32 and 0.35 eV (Figure 2a), which could be attributed to transitions from different sub-valence bands [25]. We denoted the range of the stronger resonance with IR photon energy of ~0.32–0.36 eV as zone I (shaded in blue) and the remaining as zone II (shaded in red) (Figure 2a).

We then acquired TR-SFG spectra from the GaAs (100) sample. The typical SF response retrieved by the CCD camera with the Galvo mirror in action is shown in Figure 2b. Each spectrum was averaged from 10 sets of measurements, and each set had an acquisition time of ~4 s. In the absence of pump excitation, the static SF spectra of GaAs collected from ROI 1 and 2 on the CCD chip were identical as expected (upper panel in Figure 2b). With the NIR pump on, the pump-on SF spectrum (ROI 1) became deviated from the pump-off spectrum (ROI 2), as shown in the bottom panel in Figure 2b (at *t* = 150 fs, with the pump fluence being ~2.2 mJ/cm^2^). We then extracted the ∆I/I(ω, t) spectra versus the time delay *t* using Equation (3), as presented in Figure 2c in the form of a 2D map. After the photoexcitation, the transient SFG intensity within zone I dropped immediately, while that within zone II increased, showing contrasting behaviors as observed in previous pump-probe studies on GaAs [25,26]. Details of the dynamics can be found in the literature and are not covered in the present study.

We further extracted the relaxation dynamics by integrating ∆I/I(ω, t) within zone I and zone II as functions of the time delay *t*, presented in Figure 2d. For zone I (upper panel), the dynamics was characterized by an initial ultrafast decrement that can be attributed to the ground state depletion, followed by a subsequent recovery [25,26]. The rapid onset of the kinetics upon pump excitation could be fitted by the convolution of the IRF with an exponential decay function [27]. Here, we used a Gaussian function for the IRF and extracted a HWHM of ~65 fs via fitting (gray dashed line in top panel in Figure 2d). The subsequent recovery of the signal was fitted by a double exponential function, consisting of a fast component of ~0.12 ± 0.02 ps and a slow component of ~16 ± 7 ps, which could be attributed to the intraband recombination and thermalization processes [25,26]. For zone II, the dynamics exhibited a sharp rise, possibly due to excited state absorption and/or the spectral broadening of the SFG resonance upon photoexcitation as can be seen in the lower panel of Figure 2b [28]. The dynamics was then followed by a fast decay on a time scale of ~0.71 ± 0.07 ps and a slower decay on a time scale of ~30 ± 3 ps (lower panel in Figure 2d). The above results demonstrated that our experimental scheme could readily obtain the ultrafast dynamics SF responses, with a time resolution better than 100 fs.

### 3.2. TR-SFG Results of TMDC/Organic Thin Film Interfaces

We applied the TR-SFG technique to study hybrid TMDC/organic molecule interfaces, using the MoS_2_/PMMA and WS_2_/PMMA interfaces as prototype systems. White-light optical images of the MoS_2_/PMMA and WS_2_/PMMA samples are shown as insets in Figure 3a. Figure 3a also presents the photoluminescence (PL) spectra (under 532 nm CW excitation) of the MoS_2_ and WS_2_ monolayers at room temperature. The MoS_2_ monolayer showed a PL peak centered at ~1.90 eV corresponding to the X_A_ exciton, and the WS_2_ monolayer showed the X_A_ exciton PL peak at ~2.02 eV [29]. To characterize the PMMA films, we acquired FTIR spectra of the MoS_2_/PMMA and WS_2_/PMMA samples, as shown in Figure 3b. The lower transmittances below 2400 cm^−1^ and above 3600 cm^−1^ were due to the absorption of the silica substrate. Within the C–H stretching vibration region between 2700 and 3000 cm^−1^, three major adsorption peaks at 2834, 2950, and 2991 cm^−1^ were observed in both samples, which could be assigned to the symmetric stretching vibration (*ν*_s_) of the main-chain methylene (CH_2_) groups, symmetric stretching vibration from ester methyl group [(O)CH_3_], and asymmetric (*ν*_as_) stretching vibration from both ester and alpha methyl groups [(C)CH_3_] on side chains [30]. In addition, we found the transmittance from both samples had similar magnitudes, proving that the PMMA films had nearly the same thickness.

We then studied the SF spectra of the PMMA interfaces. For reference, we first took the spectrum from a PMMA thin film deposited on silica substrate without TMDC layers, as presented in Figure 3c (dotted curve, magnified by 100 times). The spectrum was dominated by a single peak near 2951 cm^−1^, similar to that reported in the literature [31]. According to Ref. [31], the response is primarily from the PMMA/air interface, and the dominance of the *ν*_s_-(O)CH_3_ mode indicates that the *C*_3v_ symmetric axes of (O)CH_3_ groups align preferentially with the surface normal.

The static SF spectra of MoS_2_/PMMA and WS_2_/PMMA are shown in Figure 3c and d, respectively. Since MoS_2_ and WS_2_ both belong to the non-centrosymmetric *D_3h_* point group, they would also exhibit SF responses [32]. In the case of MoS_2_/PMMA, the central photon energy of the SF signal was at 1.91 eV, which aligned with the X_A_ exciton at 1.90 eV [29]. The intense SF response showcased the large exciton–photon coupling strength in MoS_2_ [33]. On top of that, three distinct dips were observed at the C–H stretching vibrational frequencies of PMMA (Figure 3c). Here, the resonances exhibited as dips instead of peaks due to the interference effect between the broad exciton resonance and sharp molecular resonances [13,34]. Meanwhile, instead of the dominance of the *ν*_s_-(O)CH_3_ mode observed at the air interface as in Ref. [31], the relative strengths of *ν*_as_-CH_3_ (2989 cm^−1^) modes became much greater for MoS_2_/PMMA. This indicated that the side chains aligned preferentially toward the MoS_2_ monolayer, instead of toward the surface normal at the air interface [31]. The above result showed that SFG could capture both the vibrational signatures of organic molecules and the electronic transitions of TMDCs at such hybrid interfaces.

Notably, the strength of PMMA resonant “dips” is much greater than that of PMMA signals on silica by orders of magnitude. We calculated the Fresnel coefficients at both PMMA/air and substrate/PMMA interfaces using the modified matrix formalism [35] (see Appendix A). For a PMMA film thickness of ~250 nm, the Fresnel coefficients at both interfaces were only different by about 10%. To further separate the interference effect between excitonic and vibrational transitions, we fitted the entire spectrum using Equations (1) and (2) and extracted the resonant amplitudes of PMMA vibrational modes. The X_A_ exciton of MoS_2_ and C–H stretching modes of PMMA were described using Lorentzian line shapes of different parameters (see details in Appendix A). After removing the signal from the X_A_ exciton of MoS_2_, we reconstructed the SF spectrum of PMMA alone, as plotted in Figure 3c (shaded in red). It was seen that apart from the interference effect, the SFG amplitudes of PMMA C–H stretching vibrational modes, in contact with MoS_2_, could be enhanced by over ~10 times that on silica (see Appendix A). Hence, in the presence of the MoS_2_ monolayer, the spectrum was dominated by the enhanced response from the MoS_2_/PMMA interface, which overwhelmed that from the PMMA/air surface.

Despite the similarity in FTIR spectra, the static SF spectrum of WS_2_/PMMA (Figure 3d) was quite different from that of MoS_2_/PMMA. In the case of WS_2_/PMMA, the SF spectra only showed a nearly featureless signal within this range, increasing steadily toward higher frequency in accordance with the X_A_ exciton energy of WS_2_ being 2.02 eV (Figure 3d). Meanwhile, the vibrational signatures from PMMA became barely discernible. Spectra from different locations across the samples consistently showed the prominent PMMA signals on MoS_2_ but weak signals on WS_2_ (see Appendix A). We again extracted the PMMA C–H vibrational resonances via fitting (see Appendix A) and reconstructed the SF spectrum from PMMA alone as plotted in Figure 3d (shaded in blue), which turned out to be comparable to that of silica/PMMA but much weaker than that of MoS_2_/PMMA (Figure 3c). After excluding the IR attenuation and interference effects, we found the SF amplitudes of PMMA vibrational modes on MoS_2_ were still about ten times greater than those on WS_2_ and silica (see details in Appendix A). Such a contrast between the two TMDC/PMMA interfaces strongly suggested that the excitonic excitation in MoS_2_ was responsible for the significant enhancement of PMMA vibrational signals. Upon resonance, the X_A_ exciton in MoS_2_ could largely enhance the local electric field due to its giant exciton oscillator strength, causing the hyperpolarization (oscillating at the sum frequency) of nearby C–H vibrations to enhance (schematically sketched in Figure 3e). In contrast, systems lacking the exciton resonance, such as WS_2_/PMMA, would show negligible enhancement effects on the molecular SF signals. Recently, Ling and coworkers reported an excitonic enhancement of Raman scattering signals on 2D/organic hybrid structures [36]. Our results suggest that such an enhancement could also occur in coherent optical transitions such as SFG.

We now utilize TR-SFG to further inspect this phenomenon. The ∆I/I(ω, t) spectra from MoS_2_/PMMA and WS_2_/PMMA interfaces are displayed in Figure 4a,b in the form of 2D maps, acquired at a pump fluence of ~4.4 mJ/cm^2^. Representative spectra at various time delays are shown in Figure 4c,d, which are offset vertically for clarity. At the spectral edges, the SNR became poor because of diminishing $I\left(\omega ,\;t\right)$ (used as denominators), so we only present the signal in the IR frequency range between 2700 and 3700 cm^−1^. We further extracted the time-dependent dynamic responses of MoS_2_, WS_2_, and PMMA SF signals from Figure 4a,b, as presented in Figure 4e–g. The data presented for MoS_2_ and WS_2_ were ∆I/I(ω, t) integrated in the entire spectral range, while those for PMMA were obtained via fitting each of the SF spectra at various values of *t*.

In the case of MoS_2_ (Figure 4a,c,e), the ∆I/I(ω, t) spectra between 1.88 and 2.00 eV showed an intense negative response from *t* ~ −0.2 to +0.2 ps upon excitation (Figure 4c). Based on the static SF spectrum (Figure 3c), we attributed the initial negative band to the bleaching caused by the pump beam. In the case of WS_2_ (Figure 4b,d,f), the ∆I/I(ω, t) also showed a rapid drop near *t* = 0 but more negative toward the higher frequency, in accordance with the X_A_ exciton energy being 2.02 eV for WS_2_, which was higher than that of MoS_2_. Soon after *t* = 0, the TMDC signals both started to recover. We fitted the relaxation processes using a multi-exponential decay function and extracted the characteristic time constants for MoS_2_ to be ~0.15 ± 0.04 ps, 12 ± 3 ps, and 200 ± 25 ps (Figure 4e) and those for WS_2_ to be ~0.25 ± 0.06 ps, 13 ± 2ps, and 167 ± 28 ps (Figure 4f). These values agreed with typical time constants obtained for the two TMDC materials [5,8,37]. According to the literature, the fast decay time of 0.1~0.2 ps could be ascribed to the direct relaxation of excitons (combination of radiative and nonradiative) and the slower decay time on the order of 10 ps related to carrier-phonon scattering or slowing carrier recombination from dark or defect states. The slowest component over hundreds of picoseconds was consistent with previously reported values for the direct recombination of excitons and/or slower carrier capture processes [38].

Meanwhile, we also observed a pronounced dynamic response of the PMMA vibrational spectrum on MoS_2_ (Figure 4a,c). The PMMA signature on WS_2_ remained insignificant as on static spectra (Figure 4b,d). Again, to disentangle the interference effect with the MoS_2_ X_A_ exciton, we fitted all SF spectra from the MoS_2_/PMMA interface at various values of *t* (see Appendix A) and singled out the PMMA SF response as performed for Figure 3c. Since the two major C–H vibrational bands exhibited similar dynamics (see Appendix A), we plotted the sum of the resonance amplitudes of all PMMA vibrational modes versus *t*, as shown in Figure 4g. Interestingly, the PMMA signal experienced an increment first between *t* ~ −1 and −0.3 ps, then a recovery process resembling those of TMDCs. Previously, we have attributed the enhancement of PMMA vibrational SF signals to the local field effect arising from the MoS_2_ X_A_ exciton. At *t* ~ −1 to 0 ps, although the pump beam did not fully overlap with the IR beam, it could already excite excitons via other processes such as multi-photon processes [39]. Although such excitons by pump alone were not coherent with the PMMA SF emission, the latter might still experience an enhancement effect to some extent. After *t* = 0, the PMMA signal mostly sensed the modulation from MoS_2_ X_A_ excitons generated by the probe beams, which resonated coherently with the PMMA SF response. We further noted that the PMMA signal exhibited oscillatory behavior between *t* ~ 0 and 1 ps, which possibly reflected the interference effect from different contributions (Figure 4g). Further investigations would be necessary to explore this process with more details.

## 4. Conclusions

In summary, we implemented TR-SFG spectroscopy at TMDC/organic interfaces and demonstrated the effectiveness of this technique in probing excitonic and vibrational dynamics at these buried interfaces. Our experimental setup employed a Bragg grating for generating both the narrowband probe beam and the ultrafast pump pulse for the TR-SFG experiment, along with a synchronized beam chopper–Galvo mirror scheme to achieve real-time normalization of pump-on and pump-off SF spectra. When applied to the TMDC/PMMA interfaces, we obtained orientational information about the PMMA side chains in contact with the TMDC monolayers and observed a significant enhancement of the PMMA molecular SF response (for about 10 times in amplitude) that resonated with the TMDC excitonic transitions. Combined with ultrafast dynamics results, we attributed this phenomenon to the local field enhancement from coherent excitation of TMDC excitons adjacent to the organic molecules. We anticipate further investigation of this excitonic–vibrational interaction across TMDC/molecular interfaces, which may help elucidate coupling mechanisms in such hybrid structures and lead to potential applications such as surface-sensitized molecular sensing and so on.

## Figures and Tables

**Figure 1 nanomaterials-14-01892-f001:**
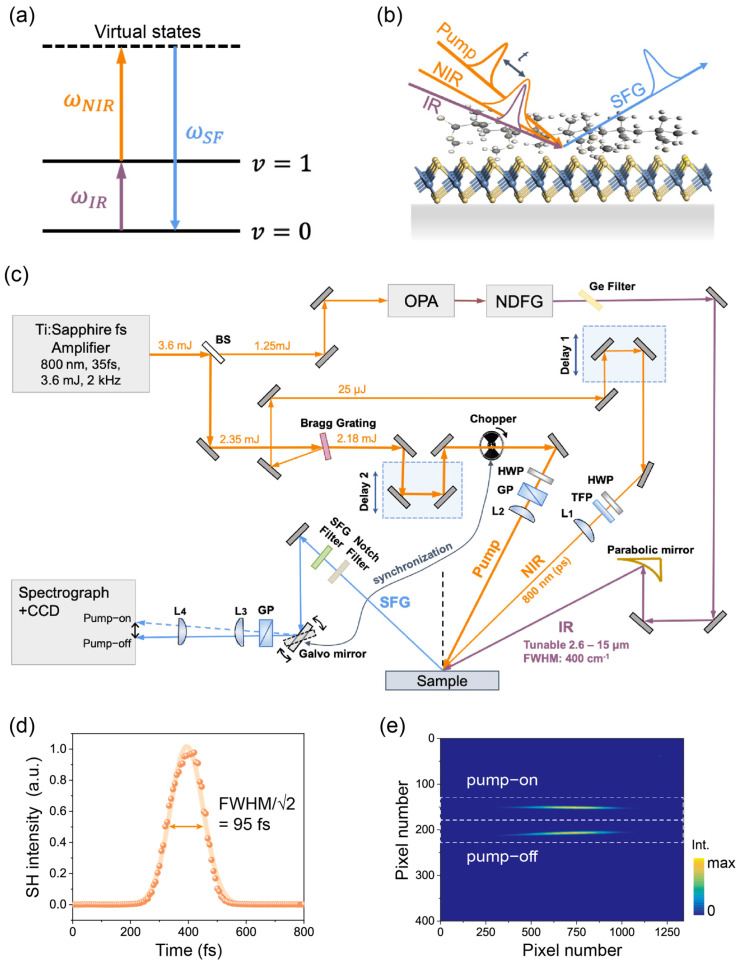
(**a**) Energy diagram for the SFG process in this study: an infrared (IR) photon excites the system from the ground state (ν=0) to a vibrational excited state (ν=1) upconverted by the near-infrared (NIR) photon to an upper state, and then emits an SF photon. (**b**) Schematics of the TR-SFG experimental geometry. *t* is the time delay of the pump and probe pulses. (**c**) Schematics of the laser experimental setup. Notations: BS, beam splitters; OPA, optical parametric amplifier; NDFG, non-collinear difference frequency generation; HWP, half-wave plate; TFP, thin-film polarizer; GP, Glan-Laser calcite polarizer; parabolic mirror, off-axis parabolic mirrors with protected gold coating; L1–L4, focusing lenses. (**d**) SHG cross-correlation of the pump pulse fitted with a Gaussian function (orange curve). (**e**) Typical image of the SF signal on CCD camera with the chopper–Galvo system in action. The upper and lower traces corresponded to the pump-on and pump-off signals, respectively. The white dashed lines indicate the regions of interest (ROIs) for retrieving the pump-on and pump-off SF spectra.

**Figure 2 nanomaterials-14-01892-f002:**
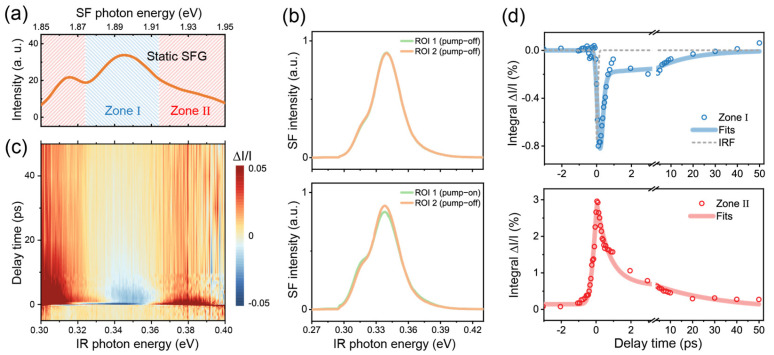
(**a**) Static SF spectra of the GaAs (100) sample. (**b**) SF spectra retrieved from different ROIs on the CCD camera with the Galvo mirror in action. The top panel shows the spectra with the pump beam blocked; the bottom panel shows the spectra with the pump beam set on, at a time delay of 150 fs. ROI 1 and 2 correspond to those marked in Figure 1e. (**c**) A 2D pseudo-color map of TR-SFG spectra from GaAs (100). (**d**) TR kinetic profiles probed at zone I (top panel) and zone II (bottom panel) for the GaAs (100) surface. In the top panel, the instrument response function (IRF) is shown by the dashed gray trace.

**Figure 3 nanomaterials-14-01892-f003:**
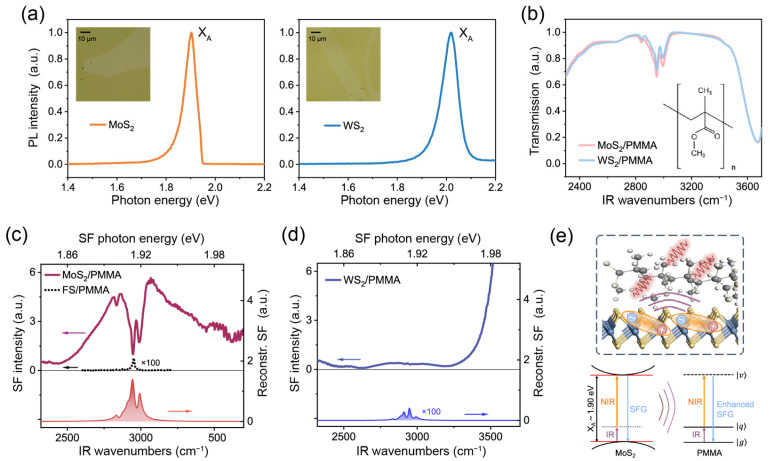
(**a**) PL spectra of MoS_2_ (left panel) and WS_2_ (right panel) monolayers excited by 532 nm CW laser (the asymmetry in the MoS_2_ PL line shape came from the long-pass filter used in measurement). Insets are the white-light optical images of the TMDC/PMMA samples. (**b**) FTIR spectra of the MoS_2_/PMMA (red) and WS_2_/PMMA (blue) samples. Inset shows the molecular structure of PMMA. (**c**,**d**) Static SF spectra of the MoS_2_/PMMA and WS_2_/PMMA interfaces (solid curves). Curves with shaded areas present the reconstructed SF spectra of PMMA via fitting. Spectra are vertically offset for clarity. The dotted curve in (**c**) presents the SF spectrum of the silica/PMMA sample for reference. (**e**) Schematic diagram of the exciton induced local field enhancement for the vibrational SF signal at the MoS_2_/PMMA interfaces.

**Figure 4 nanomaterials-14-01892-f004:**
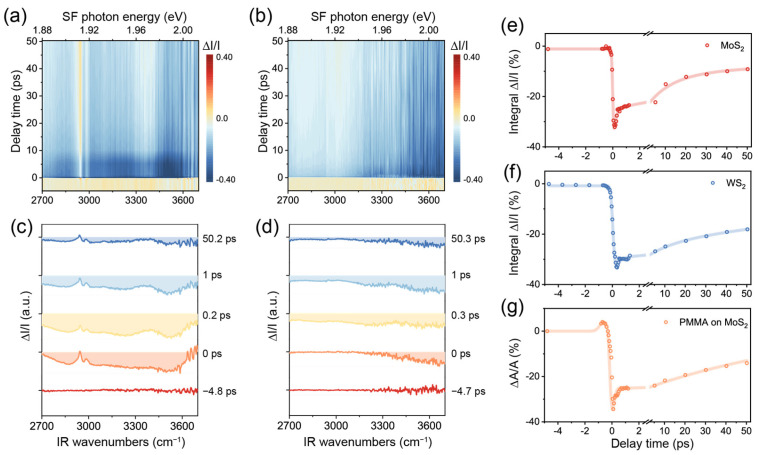
(**a**,**b**) The 2D maps of TR-SFG spectra from the MoS_2_/PMMA and WS_2_/PMMA interfaces. (**c**,**d**) TR-SFG spectra at various time delays from the MoS_2_/PMMA and WS_2_/PMMA interfaces. (**e**,**f**) Kinetic traces of integrated TR-SFG signal of the MoS_2_/PMMA and WS_2_/PMMA interfaces. (**g**) Kinetic trace of the PMMA vibrational SF signal at the MoS_2_/PMMA interface.

## Data Availability

The data that support the findings of this study are available within the article and upon request.

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
