# Peer review of "Excitonic-Vibrational Interaction at 2D Material/Organic Molecule Interfaces Studied by Time-Resolved Sum Frequency Generation"

_nanomaterials, 2024, doi:10.3390/nano14231892_

Round 1

Reviewer 1 Report

Comments and Suggestions for Authors

This paper by H. Chen et al. describe electronic-vibrational coupling at 2-dimensional material (TMDC)/PMMA interfaces via time-resolved SFG. Applying this technique to the TMDC/PMMA interfaces yielded structural information regarding PMMA side chains and dynamic responses of both PMMA vibrational modes and TMDC excitonic transitions. They also observed a prominent local field enhancement effect between the PMMA vibrational SF signal and TMDC excitonic transition when in resonance.

The interpretation of this paper is, however, qualitative only and it seems to be lacking in quantitative analysis. More detailed and quantitative analyses are needed on the points listed below.

1.       Although the thickness of the PMMA layer is estimated to be 250 nm from IR, it is necessary to quantitatively evaluate how much the effect of interference due to multiple reflections of SFG in this PMMA layer thickness is reflected in the decrease in SFG intensities of the PMMA on TMDC. To determine the impact of the film thickness, the increase or decrease in signal intensities for the PMMA film thickness can be obtained experimentally and by calculating the Fresnel coefficient considering multiple reflections, after varying the PMMA film thickness.

2.       It seems odd that if 250 nm PMMA was stacked on TMDC, the SFG signals from the topmost surface of PMMA would completely disappear, even if there was an impact from the exciton of TMDC. The authors should discuss in more detail the reasons for the disappearance of the SFG signals from the top surface.

3.       No description of the relaxation time of time-resolved VSFG can be found for a longer relaxation process.

4.       This experiment is performed using VSFG, but no specific mention of vibrational relaxation is found. Wouldn't it be sufficient to do this with time-resolved SHG?

Reviewer 2 Report

Comments and Suggestions for Authors

The manuscript with the title “Excitonic-Vibronic Interaction at 2D Material/Organic Molecule Interfaces Studied by Time-Resolved Sum Frequency Generation” studies exciting interface properties with a very complicated setup. The authors used three different laser beams to excite, pump, and probe the exciton-vibronic interaction and get the time-resolved kinetics. Generating the sum frequency of two different energy laser beams is very interesting on TMD/PMMA interface and combining the excitonic behavior for the enhancement of the nonlinear processes. PMMA is a good probe there since it has really clear vibrational bands in the spectral region where MoS2 can enhance the SFG  

The topic itself is very interesting, and the method they used to study optical properties is advanced. However, some points require clarification and maybe some corrections. I would suggest publishing the manuscript after major revision, which should answer the following comments.

1)      On page 3, where they describe the sample preparation, they can add information there that they used GaAs as reference material. It was very confusing when it read from top to bottom since GaAs has nothing to do with the actual samples. It is just for testing the setup and proving that the optical setup measures the SFG as it should.

2)      On line 134, they mention that the transmitted pump pulse energy is 2.2 mJ. What does this number correspond to? Is it just the transmitted pump beam through the Bragg grating? Also, how is this Bragg grating working? If I understand correctly, it is reflecting a very narrow bandwidth of the spectrum from the pump beam as an NIR probe beam. Does that mean that that part of the spectrum is missing in the transmitted beam? Can authors comment on this?

3)      I am a bit confused about seeing two delay lines on the optical setup. Can authors comment on which one they are delaying? To me, either probe or pump should be delayed relative to the other, but I am confused that both arms have a delay stage in their case.

4)      In line 206, the authors discuss zones 1 and 2. I agree that zone 1 has an enhanced peak, corresponding to A exciton of the MoS2. It is brilliant that they used the resonant effect to enhance SFG. Can authors comment on the other peak in the zone 2 left side? Does it correspond to any resonance as well?

5)      On line 227, I guess it is a typo that the authors are referring to 2d, not to 4d.

6)      In Figure 3a, they show the MoS2 PL spectrum, which looks asymmetric. Can the authors comment on why the spectrum does not look like Lorentzian, as is the case in WS2?

7)      In Figure 3 insets, bright field images of the flakes are given. I am concerned that monolayer flakes look very small compared to IR and NIR beams. I wonder how this size difference affects the pump-probe measurements, and is this important if we compare MoS2 with WS2? Since there is some conclusion drawn by the authors saying the exciton energy overlapping is the key to the enhancement of SFG (in principle, I agree, there is strong evidence for that), I think authors should clarify and provide additional data to show that small flakes do not affect the conclusion.

8)      Can authors clarify, what they mean by saying interference effect on line 269. To me, I think there is enhancement of the nonlinear effect (SFG) due to A exciton overlapping (i.e. resonance effect) but I think that is what they see as a broad peak on the spectral range. Further, PMMA has very strong vibrational peaks on the region and PMMA absorbs. I would not call this effect as enhancement of PMMA. I would say it is happening in the same manner for the WS2 case, but since the enhancement is very small at the beginning, the absorption of the PMMA looks very minor too. I think the authors should provide a quantitative analysis with enhancement factors and relative PMMA absorption in that region. In the current version (lines 294-295), it is very hard to conclude how much PMMA effect is visible. This part should be improved significantly and revised to bring some clarity.    

9)      On line 270, they mentioned the vibration energy of -(O)CH3 in air, and maybe this should be added in SI to compare the spectra.

10)  I don’t agree with calling the effect an “enhancement effect to the molecular SF.” As I mentioned above, this doesn’t seem to be an enhancement of PMMA SF.

11)  Figure 4 e and 4f show kinetics for MoS2 and WS2. Since the probe beam is spectrally very narrow, how should we interpret which effect we are probing in the 4c and 4d? Can authors clarify this point?

12)  General comments about the abstract and conclusion. I think the main message is not clear and the enhancement of the SFG is not quantified. This needs to be improved.

13)  I am also unsure about the relevance of reference 35 in your case. Authors should double-check that the references are correctly cited.

Round 2

Reviewer 1 Report

Comments and Suggestions for Authors

This revised version of the manuscript is significantly improved, and the authors tried to address all the points raised by the reviewers. Thus the manuscript in the current version is suitable for publication in Nanomaterials as is.

Reviewer 2 Report

Comments and Suggestions for Authors

Thank you very much for the detailed explanation and the address in all my comments. I am happy with the improvement of the manuscript and message of the study is clearer to me.  I have only one comment about the quantitative measurements and the enhancement comparison. I understand that the authors are planning to study this part in the future, but it is more valuable to put exact numbers in the manuscript to speak about the exciton contribution to the SFG. I appreciate the effort the authors made in the revision (by adding ~10 times and ~100 times enhancements). Hoping to see the future studies of this interesting topic.